# Augmenting a ResNet + BiLSTM Deep Learning Model with Clinical Mobility Data Helps Outperform a Heuristic Frequency-Based Model for Walking Bout Segmentation

**DOI:** 10.3390/s25206318

**Published:** 2025-10-13

**Authors:** Matthew C. Ruder, Vincenzo E. Di Bacco, Kushang Patel, Rong Zheng, Kim Madden, Anthony Adili, Dylan Kobsar

**Affiliations:** 1Department of Kinesiology, McMaster University, Hamilton, ON L8S 4L8, Canada; dibaccov@mcmaster.ca (V.E.D.B.); kobsard@mcmaster.ca (D.K.); 2Department of Computing and Software, McMaster University, Hamilton, ON L8S 4L8, Canada; patek139@mcmaster.ca (K.P.); rzheng@mcmaster.ca (R.Z.); 3Department of Surgery, McMaster University, Hamilton, ON L8S 4L8, Canada; maddenk@mcmaster.ca (K.M.); adilia@mcmaster.ca (A.A.); 4Research Institute of St. Joe’s, Hamilton, ON L8N 4A6, Canada

**Keywords:** deep learning, gait recognition, human activity recognition, inertial sensors, osteoarthritis

## Abstract

Wearable sensors have become valuable tools for assessing gait in both laboratory and free-living environments. However, detection of walking in free-living environments remains challenging, especially in clinical populations. Machine learning models may offer more robust gait identification, but most are trained on healthy participants, limiting their generalizability to other populations. To extend a previously validated machine learning model, an updated model was trained using an open dataset (PAMAP2), before progressively including training datasets with additional healthy participants and a clinical osteoarthritis population. The performance of the model in identifying walking was also evaluated using a frequency-based gait detection algorithm. The results showed that the model trained with all three datasets performed best in terms of activity classification, ultimately achieving a high accuracy of 96% on held-out test data. The model generally performed on par with the heuristic, frequency-based method for walking bout identification. However, for patients with slower gait speeds (<0.8 m/s), the machine learning model maintained high recall (>0.89), while the heuristic method performed poorly, with recall as low as 0.38. This study demonstrates the enhancement of existing model architectures by training with diverse datasets, highlighting the importance of dataset diversity when developing more robust models for clinical applications.

## 1. Introduction

Wearable inertial measurement unit (IMU) systems have emerged as powerful, cost-effective tools for evaluating walking gait in clinical contexts. These evaluations are essential for understanding disease status, progression, and response to treatments [1,2,3]. In knee osteoarthritis (OA), one of the most prevalent musculoskeletal conditions [4], IMUs have been used to assess patient function [5,6], track disease progression [7], and monitor rehabilitation response [8]. While these assessments can be invaluable in clinical environments, free-living daily life offers the potential for deeper insights into functional capacity, disease state, or response to treatment that may not be captured in controlled settings [9,10]. However, the analysis of free-living IMU data presents significant challenges due to the large volume of data across various activities, many of which may not be of interest. Accurately identifying activities, or simply when a participant is walking, remains a key hurdle in translating IMU technology for real-world applications outside of controlled laboratory environments.

Traditional heuristic or rule-based methods are often used to identify periods of walking, referred to as walking bouts. One of the simplest methods is identifying peak acceleration impacts that exceed a certain threshold, which correspond to heel strikes during walking [11]. While this method is easy to implement, it lacks robustness and is poorly suited for free-living data due the variability, not only between individuals’ gait but also within an individual’s gait in uncontrolled environments [10]. Alternatively, the cyclical nature of walking gait can be effectively incorporated into walking bout identifier algorithms by examining the signal’s frequency content. For example, the harmonic frequencies of the IMU signals can be analyzed using a fast Fourier transform (FFT) to identify walking behavior by detecting harmonics that are indicative of gait [12]. This approach has demonstrated strong performance in laboratory settings (sensitivity = 0.98), with only minimal drop-off when deployed on semi-structured, unsupervised, and remotely collected validation data (sensitivity = 0.97). While this improvement over simple peak heuristics is promising, there are still challenges related to thresholding estimates for harmonics, which can limit the generalizability and robustness for walking classification. Participants also generally walk slower in free-living assessments, which may complicate gait detection in populations with altered gait that already move slower than healthy populations [10,13]. Therefore, developing gait identification algorithms that are robust and agnostic to these variations must be accounted for before deployment on free-living data.

Machine learning (ML) methods have emerged alongside wearable IMUs and offer improved predictive power over traditional heuristic models. Techniques such as support vector machines [14] and Hidden Markov Models [15] have been used to identify walking bouts using handcrafted features from three-dimensional acceleration and angular velocity data. While effective, these methods rely on feature engineering and may not capture the full complexity of human movement. More advanced deep learning approaches, such as Convolutional Neural Networks, can automatically extract relevant features from raw IMU data, enabling more effective gait analysis [16]. However, given the temporal dependence of human movement signals, utilizing a deep learning model that also integrates the temporal structure of the signal can be critical to identifying walking bouts in free-living data.

Long short-term memory (LSTM) networks, a type of recurrent neural network, are deep learning models designed to capture long-term dependencies in time-series data. These advanced networks excel in tasks where temporal information is essential, such as sequence prediction or activity recognition, because they can retain information over long sequences and mitigate the vanishing gradient problem often encountered in traditional recurrent neural networks. A recent study by Li and Wang tested the use of this model to identify activities, including walking, from an open dataset (PAMAP2) [17]. The architecture consists of two main components: (1) the residual convolutional 2D block, which extracts features from a window of the IMU data, and (2) the bidirectional LSTM network, which captures long-range dependencies and reduces information loss in sequential data. This model’s architecture is particularly intriguing because it has demonstrated strong performance across various datasets, ranging from those with multiple sensor placements to a single sensor positioned just below the knee. Specifically, the authors found that the model identified activities with over 95% accuracy when trained independently across three datasets: the full PAMAP2 dataset, another open dataset (WISDM), and a lab-created dataset with a single proximally placed sensor on the shank. While these findings are promising, they share limitations common to many models, as highlighted in a recent review [18]. Specifically, the models were trained and validated exclusively with healthy participants, with no evaluation on individuals with altered gait, limiting the generalizability of the findings. Moreover, these results were obtained without a specific, separate test set and were not compared to traditional heuristics, making it difficult to assess the model’s performance relative to simpler, more established methods.

Rather than developing a new ML architecture, this study extends Li and Wang’s [17] approach to create a clinically viable system that uses a single lower-limb sensor capable of identifying activities in both healthy individuals and those with knee OA, with a particular focus on walking bouts, due to their significance in these populations. A model architecture similar to that of Li and Wang was employed. While one of their base models was trained only on PAMAP2 data, this study first retrained that particular base model and then augmented training with additional data from both healthy individuals and patients with knee OA, a clinical population seldom incorporated into activity recognition models. The integration of clinical OA data represents a novel step toward advancing activity recognition approaches in patient populations. Additionally, the performance of this trained deep learning model was evaluated against a common heuristic gait detection algorithm [12] to compare ML performance with a heuristic approach, using a more comprehensive test set spanning a variety of gait speeds. Specifically, we hypothesized that (i) the deep learning model, when trained with additional healthy and clinical data, would outperform the base open dataset model on a clinical test set; (ii) it would outperform the heuristic gait detection algorithm; and (iii) this advantage would be particularly evident in individuals with lower gait speeds. In this way, this study aims to develop a clinical gait detection model using only a single sensor located on the shank.

## 2. Materials and Methods

### 2.1. Description of Datasets

The data used in the current study were sourced from three distinct datasets: (a) the PAMAP2 dataset [19], (b) a new healthy adult dataset, and (c) a new clinical dataset of adults with knee and hip OA. Each dataset contributed to the development, validation, and testing of the model, with data obtained from the IMU placed at varying locations on the shank. The PAMAP2 dataset, which includes a variety of activities and sensor placements (chest, wrist, and ankle), served as the base dataset for model development. In this application, only the ankle sensor data were used to simplify the sensor array and facilitate the extraction of meaningful gait metrics from OA patients using this location, extending the scope of the current study [20]. The inclusion of (b) a secondary healthy dataset, in which IMUs were placed more proximally on the shank, was intended to promote the development of a model that is more robust to variations in sensor placement along the shank (Figure 1). Next, the limited number of healthy participants in the base dataset may restrict its generalizability to older adults with gait impairments. To address this, (c) a clinical dataset comprised of adults with knee and hip OA, collected approximately two weeks prior to joint replacement surgery, was included to increase population diversity. This dataset represents end-stage OA and provides valuable data on individuals exhibiting a range of gait speeds and condition-related gait alterations [13]

Specifically, as noted, the participants, the IMU sensors used, sensor placement, and activities completed varied across the datasets. The PAMAP2 dataset [19] included nine participants (8M, 1F; 27.2 ± 3.3 y; 179.4 ± 8.4 cm; 80.9 ± 10.3 kg), wearing multiple IMUs sampled at 100 Hz while performing 12–18 activities; for the current study, only the dominant ankle IMU was used, yielding 13 activity labels before preprocessing (with Subject 109 excluded). A total of 500 samples were removed from the start and end of each activity to exclude transitional periods and labeling inaccuracies. The healthy lab dataset consisted of 14 healthy participants (10M, 4F; 25.0 ± 4.4 y; 180.1 ± 9.1 cm; 77.1 ± 13.4 kg), who wore bilateral tibial-mounted IMUs (IMeasureU Blue Trident, 1600 Hz, downsampled to 100 Hz) during walking, static, and non-gait ambulation tasks; left/right data were concatenated after non-labeled data were removed. Finally, the clinical dataset included 32 older adults with OA awaiting knee (n = 25) or hip (n = 7) arthroplasty (12M, 20F; 65.7 ± 7.8 yr; 168.0 ± 9.9 cm; 92.3 ± 23.9 kg), who wore bilateral tibial-mounted IMUs (Axivity AX6, 100 Hz) during one week of free-living and in-clinic assessments; only quiet standing and 60 s self-selected walking tasks during the in-clinic assessments were labeled, with gait speeds stratified as slow (<0.8 m/s), average (0.8–1.2 m/s), or fast (>1.2 m/s). Full descriptions of the datasets and activity labels, along with the full protocols for the healthy and clinical datasets, are provided in Appendix A.

### 2.2. Data Preprocessing

All data processing was performed in Python 3.9. Identical procedures were applied across the PAMAP2, healthy adult, and clinical OA datasets. Only the acceleration and gyroscope signals were utilized given that these were consistently available across all three datasets.

First, sensor signals were calibrated to align with the shank coordinate system. Following the procedure described by Mihy et al., quiet standing trials were used to orient the gravity vector vertically, and a subsequent rotation around the vertical axis was applied to a section of walking data to align it to the anteroposterior direction [21]. Units were then standardized as needed to express acceleration in meters per second squared (m/s^2^) and angular velocity in degrees per second (deg/s).

Activity labels were harmonized across datasets into seven categories (static, walking, running, cycling, stair ascent, stair descent, and other activities), as outlined in Appendix A. The “static” label included all non-movement activities (e.g., sitting, standing, or lying), while “other” captured activities of daily living that are not representative of gait behavior, such as vacuuming or house cleaning, which would not be completely static but would also not constitute true gait. The “other” category was created to limit the scope of the model, as the activities in this category had limited samples but could also appear gait-like in their signal presentation, which could ultimately improve model performance. Overall, the included activities were selected to be representative of potential activities in the clinical population and utilized as needed. Full details are provided in Appendix A. For all datasets, unlabeled and unused activity-labeled data were removed.

Following event relabeling, sensor data was segmented into n-second sliding windows (length n · sampling frequency, 50% overlap). For static-labeled windows, the root mean squared acceleration was required to be less than 1 m/s^2^. While the original Li and Wang paper uses a one-second window (window_sz = 100), a leave-one-subject-out analysis found minimal changes in accuracy up to five seconds (window_sz = 500). Given that the target population for this model are clinical patients who may walk slower, the window was extended to five seconds, which also aligns with the heuristic frequency-based method [12] used in the secondary analysis. Participant-specific scaling was applied using the walking section to generate a transform using the StandardScaler function (SciKit-Learn v1.6.1), which was then applied to all data for that participant.

### 2.3. ResNet + BiLSTM Model Framework

The model implemented in this study was structured identically to the architecture outlined by Li and Wang [17]. In short, there are two major components within the model: a residual network (ResNet) block and a bidirectional long short-term memory (BiLSTM) block. The ResNet is used to extract spatial features from the signal, while the BiLSTM captures the forward and backward temporal information from time sequences (Figure 2). Together, these components combine to form a framework that is capable of learning both spatial patterns and long-range temporal dependencies, which are both essential for modeling human movement.

The input to the model consisted of segmented, sequential IMU data, represented as a tensor of size (window_sz, n_channels,1). The window_sz represents the number of samples in the current sequence, which is set to 500 samples in this study, and n_channels represents the six IMU channels included (i.e., three accelerometer and three gyroscope axes).

To extract spatial features from the input data, ResNet was introduced. The ResNet is comprised of two convolutional layers and an additional convolutional shortcut connection. Each convolutional layer is composed of 32 kernels with a size of 2 × 2. The first convolutional layer employs a stride length of 2 for its convolutional window. Following this layer, a batch normalization (batch norm) layer is used to accelerate training and re-center the data before passing to the next layer. Next, ReLU is used as the activation function before passing to the second convolutional layer. Following the second convolutional layer, which has a stride length of 1, another batch norm layer is used. At this point, the shortcut connection, featuring the third convolutional layer with a stride length of 2, is added to mitigate vanishing gradients, enabling deeper networks to learn effectively. The ResNet concludes with a second ReLU activation function, followed by a dropout layer with a dropout rate of 0.5. Before being passed to the BiLSTM layer, the resulting output to this point is passed to a flatten layer, which collapses the dimensions into a 1-dimensional array. The output from the ResNet is then passed to the BiLSTM block.

While the ResNet extracts the spatial features and local patterns of the signals, the BiLSTM captures the temporal relationships of the time sequences from the output. Unlike a standard LSTM, which can only propagate information forward in time, the BiLSTM analyzes sequences in both the forward and backward temporal relationships of the signals. The BiLSTM includes both forward and backward layers, whereas a standard LSTM network would only be based on previous data. The bidirectional nature of this layer allows for better learning with respect to human activity recognition, where contextual cues from both past and future sequences can improve classification accuracy. Following the BiLSTM layer, there is an additional dropout layer with a dropout rate of 0.5.

The resulting features are passed to a fully connected dense layer that links every neuron from the previous layer, combining the learned representations and helping to prevent overfitting. The output of the data is extracted through a softmax activation layer, which finally predicts the probability of each activity. The predicted probability can then be used to predict the most likely activity for a given window.

### 2.4. Model Training and Performance Analysis

The models were trained on a desktop computer equipped with a 3.5 GHz 16-Core CPU, with 32 GB of RAM, and a graphics processor (NVIDIA GeForce RTX 2070 Super). The algorithm was implemented in Python 3.9 with TensorFlow 2.10.1, using Spyder as the integrated development environment on a 64 bit version of Windows 10. Three models were independently trained with progressive combinations of the PAMAP2, healthy, and clinical datasets, starting with only the PAMAP2 dataset, then adding the healthy dataset, and finally the clinical dataset (Table 1). Participants in each dataset were randomly divided into training, validation, and testing sets, with least 75% used for training. Validation and testing sets were randomly and evenly divided, except for the clinical dataset, where testing participants were specifically selected to ensure a range of gait speeds. The same hyperparameters as those in Li and Wang were used, except for training time. The models were trained by minimizing sparse categorical cross-entropy using the Adam optimizer. A batch size of 64 was used to train each model. Whereas Li and Wang used a training time of 80 to train each model, early stopping was used to prevent model overfitting, with a patience value of 10. The best model based on minimized loss was retained from each model training and saved as the final model.

Metrics were calculated for each final model to describe the overall performance across training, validation, and testing. For each classification, true positives (TPs), true negatives (TNs), false positives (FPs), and false negatives (FNs) were computed using functions from the the SciKit-learn package. Specifically, performance based on accuracy (1), precision (2), recall (3), and F1-score (4) was evaluated during training, validation, and testing, resulting in training, validation, and testing performance metrics, using the following equations:(1)$$Accuracy=\frac{TP+TN}{TP+TN+FP+FN}$$(2)$$Precision=\frac{TP}{TP+FP}$$(3)$$Recall=\frac{TP}{TP+FN}$$(4)$$F1-score=\frac{2*TP}{2*TP+FP+FN}$$

Test performance metrics were calculated at the conclusion of training on the held-out test sets, which comprised randomly selected participants from each dataset, as follows: one PAMAP2 participant, two healthy participants, and six clinical patients. Regardless of the included datasets used for each of the three trained models, the test set comprised all participants from all datasets to better characterize how generalizable the overall model architecture was by including additional training data.

Since this model was planned to be primarily used as a gait detection model, a secondary analysis was completed to compare performance with a heuristic model from Ullrich et al. [12]. In short, the frequency-based method uses harmonic frequencies to detect gait or non-gait by analyzing a 10 s sliding window on a given signal. For an active window (defined as the root mean square of the input signal being greater than a resting threshold), the harmonic frequency of the FFT of the window was found. If peaks were detected in at least two of the first four harmonics, then the window was deemed to represent “gait”. Conversely, if less than 2 peaks were found, then it was classified as “not gait”. The output of this method resulted in a Boolean array (i.e., true or false) indicating if gait occurred. However, because Ullrich’s study focused on foot-mounted IMUs, pilot testing with in-lab walking back and forth was performed to adapt the method data from shank-mounted IMUs. The ability of the algorithm to identify slower walking with the same signals as those used in Ullrich (i.e., vertical acceleration, acceleration norm, mediolateral angular velocity, angular velocity norm) was evaluated. In addition to the different signals, the peak prominence (i.e., how much a peak stands out relative to the signal baseline) was varied for each signal evaluation, starting with the published peak prominences in the original study. Because of the combination of slower walking speed and lower angular velocity of the shank relative to the foot, the peak prominence values were systematically reduced until accuracy began to decrease. Ultimately, in line with Ullrich’s study, the mediolateral angular velocity channel, with a peak prominence of 5, best captured instances of slower walking. The minimum signal threshold used to identify active windows from the original study was maintained at 50 deg/s.

Following training and evaluation of all models (i.e., PAMAP2 only, PAMAP2 + healthy, and PAMAP2 + healthy + clinical), the best-performing model (i.e., PAMAP2 + healthy + clinical) was tested for sensitivity in detecting instances of walking, comparing it to the heuristic frequency-based method. To achieve this, the activity predictions from the ML model were converted into a Boolean array, where each five-second segment was labeled as either “walking” or “not walking,” to match the output format of the frequency-based method. These predictions were then mapped back to the original data length. A new array, initialized to zeros, was created. As the function processed the overlapping windows, ones were added to indicate walking periods. The proportion of windows predicting walking was calculated for each sample. Finally, this array was converted into a final Boolean output, where any value greater than 0.5 was considered a “True” walking window. The analysis was performed only on labeled sections of the test sets, generating performance metrics for both the heuristic and ML models.

Additionally, as a qualitative evaluation of model performance in a free-living environment, the full data from the clinical test set was used to detect gait, using both frequency-based method and the ML model. Each participant had up to seven days of free-living data collection. Walking detection algorithms from both methods processed these data into walking bouts. Within each identified walking bout, individual strides were identified using an event detection method [22]. Mid-swing peaks were first located in the mediolateral angular velocity signal. The interval between successive mid-swing peaks was then divided in half, with the largest resultant acceleration peak in the first half classified as heel strike. Resulting strides (i.e., heel strike to heel strike) were time-normalized to 101 points to allow for ensemble averaging and comparison between the two methods within participants.

## 3. Results

### 3.1. Dataset Composition

Following preprocessing (i.e., removing unnecessary data and relabeling activities), the datasets provided varying amounts of data for training, validation, and testing. The PAMAP2 dataset provided the largest amount of data, with 50.3% of the data (1,466,715 total samples), followed by the healthy dataset, with 26.1% (760,308 total samples), and then the clinical dataset, with 23.6% (689,341 total samples). The breakdown of each activity in each dataset is provided in Table 2, both in terms of the individual datasets and as the combined datasets used for additional model training.

### 3.2. Model Training, Validation, and Testing

General model performance for each model across all datasets used in training, validation, and testing is detailed in Table 3. Model training took 46, 49, and 56 epochs before early stopping for PAMAP2 only, PAMAP2 + healthy, and PAMAP2 + healthy + clinical training datasets, respectively. In general, model performance was relatively stable in terms of training, with performance metrics (i.e., accuracy, precision, recall, and F1-score) all ranging from 0.96 to 0.98. There was a drop in model performance in the validation set, with performance metrics ranging from 0.91 to 0.93. Similar trends were seen in performance metrics when evaluating the test sets featuring held-out data from each dataset. Overall model performance on the test sets for the PAMAP2-only model achieved an accuracy of 0.85, while the PAMAP2 + healthy and PAMAP2 + healthy + clinical models both achieved an accuracy score of 0.94.

Further details of model performance with respect to performance on individual test sets from each dataset can be found in Figure 3a–c. The PAMAP2 test set performance improved slightly, with accuracy increasing from 0.92 when trained only on the PAMAP2 training data, to 0.95 and 0.94 with models trained with additional healthy and clinical data, respectively. Gait-related misclassifications decreased with additional data from the healthy and clinical datasets. Healthy test set accuracy increased substantially from the PAMAP2-only trained model to the PAMAP2 + healthy and PAMAP2 + healthy + clinical models, rising from 0.81 to 0.93 and 0.92, respectively. Clinical testing performance contained a similar trend, with accuracy increasing from the PAMAP2-only trained model to the PAMAP2 + healthy model from 0.84 to 0.95, before slightly increasing to 0.97 when clinical data was incorporated into the final PAMAP2 + healthy + clinical model.

Performance varied in part with the underlying distribution of activities across datasets. As shown in Table 4, stratifying results by activity class revealed that imbalances contributed to differences in model accuracy with higher performance on more frequent activities and comparatively lower performance on underrepresented classes.

### 3.3. Model Comparison with Heuristic Gait Detection

As the combined PAMAP2, healthy, and clinical-trained model demonstrated the best performance, this model was then used for comparison against the heuristic frequency-based method. The median performance on each test set is shown in Table 5. With the exception of the PAMAP2 testing participant, the frequency-based method performed on par with the ML-based model. The PAMAP2 testing participant only achieved an accuracy of 0.79, compared to the ML model accuracy of 0.99. The ML-based model (accuracy range: 0.96 to 0.98) performed slightly better than the frequency-based method (accuracy range: 0.87 to 0.96) on the healthy and clinical datasets. Overall, there were few notable differences.

### 3.4. Effect of Gait Speed

Further performance comparisons between the trained ML model and the heuristic frequency-based method were conducted individually on the clinical test set (Table 6). There were two patients representing each previously defined gait speed ranges: slow (0.30 and 0.75 m/s), average (1.00 and 0.99 m/s), and fast (1.32 and 1.42 m/s). The machine model performed much better than the heuristic frequency-based method on the patients with slower gait speeds, with the frequency-based method only yielding accuracies of 0.53 and 0.81, compared to the ML-based model, which achieved accuracies of 0.96 and 0.93. For patients with normal and fast gait speeds, the overall performance of both models on the healthy and clinical datasets was very similar, with accuracies ranging from 0.94 to 0.98 for the frequency-based method and from 0.97 to 0.99 for the ML model.

The average normalized mediolateral angular velocity waveforms for each participant in the clinical test set, obtained using each gait detection method, are shown in Figure 4. The mediolateral angular velocity captures the primary swing component of gait and is therefore distinctive in representing a consistent motion pattern. With the exception of Patient 3, who only had one day of data before removing the sensors, the remaining participants had at least four days of data. On average, 14,826 strides were identified with the frequency-based method compared to 25,279 strides with the ML method.

## 4. Discussion

The purpose of this study was to (a) enhance the training of a previously developed deep learning architecture, the ResNet + BiLSTM, by incorporating additional healthy and clinical participants into model training; (b) compare the sensitivity of this model to a heuristic frequency-based method; and (c) evaluate its performance across lower gait speeds. The results demonstrated that the base model, trained only on the PAMAP2 dataset, performed surprisingly well on clinical data, achieving 91% accuracy (Figure 3a). However, augmenting the PAMAP2-only model described by Li and Wang with additional clinical training data improved its performance, reaching 97% accuracy (Figure 3c). Additionally, the deep learning model generally outperformed the heuristic frequency-based method, with the greatest improvement observed at slower walking speeds in patients with end-stage OA. Although further validation is needed, particularly in free-living environments, this study presents a promising framework for augmenting existing models with additional datasets and clinical testing of the resulting models, thus enhancing gait detection models for healthy and pathological populations using wearable sensors.

In terms of model evaluation using the original architecture proposed by Li and Wang, the overall performance of the current study is comparable but modestly lower. Li and Wang reported 97% accuracy on both PAMAP2 and their own dataset, whereas this study achieved 91–93% across models. Several methodological factors likely explain these differences. Restricting PAMAP2 to clinically relevant activities (walking, cycling, and stairs) and using longer 5 s windows altered class balance and reduced training samples.

Because Li and Wang’s 70/30 split without subject separation likely introduced memory leakage, a common pitfall in ML [18], their reported performance was likely inflated. By ensuring strict separation across training, validation, and test sets, the current study offers a more realistic and honest marker of the model’s accuracy. Sensor configuration also differed. Li and Wang used all three PAMAP2 sensors, while this study relied on a single shank-mounted sensor, trading some discrimination for clinical feasibility. Augmenting with additional healthy and clinical data improved PAMAP2 accuracy to 92–95% and raised clinical test accuracy to 97%, reducing walking misclassifications (from 20% to 5%) and highlighting the value of incorporating patient data. Taken together, these comparisons clarify methodological drivers of performance differences and provide a foundation for interpreting clinical relevance in the following section.

These findings also highlight the clinical implications of the model performance. One of the main objectives of this study was to develop a trained ML model capable of accurately identifying gait bouts in a clinical OA population using only one lower-limb sensor, with performance surpassing traditional heuristic models. On the surface, the current deep learning method, augmented with clinical data, appears only slightly better than the heuristic frequency-based method (94% vs. 97%). However, when analyzed individually by gait speed, the deep learning model shows no performance drop, while the heuristic model performs poorly on slower gait speeds. Recall for the two patients with slower gait speeds was 0.38 and 0.72 for the frequency-based method, compared to 0.96 and 0.89 for the deep learning model. This ability to identify slower walking speeds is critical for both clinical and free-living applications. For instance, healthy older adults typically walk at speeds between 1.1 and 1.2 m/s [23], while patients with OA often have slower speeds, around 1.0 m/s [24,25] or slower, as seen in this study. Additionally, walking speeds in both healthy and clinical populations can be further reduced in free-living conditions outside of controlled testing environments [26,27]. Therefore, it is important for a model to accurately detect walking bouts at speeds below 1.0 m/s for real-world applications. The current model’s ability to perform well, even with a patient walking at an extremely slow speed of 0.3 m/s, demonstrates its robustness in identifying a wide range of gait types. Take together with the qualitative assessment of free-living data, which found almost twice as many strides using the ML model over the frequency-based method, the results suggest that the ML model may be effective at identifying slower gait outside of a lab environment. However, future studies should evaluate its performance in identifying clinical OA gait in free-living environments.

More broadly, these findings align with the growing body of literature applying ML to pathological gait populations. For example, a similar study incorporating gait data from individuals with Parkinson’s disease, stroke, multiple sclerosis, and chronic low back pain used a temporal convolutional neural network to identify gait events, validated against optical motion capture, at three different speeds [28]. This model performed well across both ankle and shank sensor locations, with high recall (>95%), precision (>98%), and F1-score (94%) for both initial and terminal contact events. Additionally, the previously mentioned study by Roth, et al., which used a Hidden Markov Model to identify gait in patients with Parkinson’s disease achieved high performance metrics (F1-score = 92.1%), especially for longer bouts (F1-score = 96.2%). The current study appears to largely be in line with the performance of these models in different populations, but extends this study to a broader focus on OA populations where such studies are lacking [29]. Additionally, while the current study used ResNet + BiLSTM for activity classification, the model framework has additional applications in other domains that have sequential data with both spatial and temporal dependencies, such as physiological signal modeling or fall risk detection, among others [30,31]. Overall, within the current study, the model trained with PAMAP2 + healthy + clinical data achieved high performance metrics on an OA population, which will enable future gait analyses outside of controlled laboratory environments.

While the model demonstrated strong performance, its generalizability to free-living settings remains an important consideration. The performance on clinical data in particular due to the restricted activity scope (only static and walking) may overestimate accuracy relative to the diverse environments and variable activities encountered in daily life. Given the potential of wearable devices to better capture function over traditional patient-reported outcomes, validation of this and similar models in free-living environments will be critical to establish robustness and clinical reliability [32]. With respect to gait, the model’s performance does seem to indicate an improvement over simpler heuristic models. However, a computationally intensive deep learning model like ResNet + BiLSTM may not be suitable for real-time feedback or devices with limited computational power. In these cases, the heuristic models may provide a practical implementation for identifying gait. Future research could incorporate lightweighting strategies such as model pruning, quantization, or knowledge distillation to reduce computational costs while preserving performance. Alternatively, architectures with lower complexity (e.g., CNNs) could provide the efficiency necessary for real time or wearable device implementation. For individuals with OA, the ability to better capture periods of gait could improve the ability to monitor disease progression, evaluating functional decline, and tracking rehabilitation outcomes [33,34]. Unlike conventional step counters or activity estimation that provide only crude estimates of activity, this approach captures a richer representation of gait behavior, which is more directly tied to clinical outcomes. Integration of this model into clinical workflows could provide an objective, continuous measure of mobility that complements standard patient-reported outcomes and in-clinic assessments.

This study has several notable limitations. Ideally, a dataset of older adults, both with and without pathological gait, would include a wide range of activities for model training. However, in clinical populations with gait impairment, this is not often feasible. Patients who can complete a broad range of activities are typically higher functioning and exhibit less pathological gait. As a result, using only static and walking activities for evaluation may overstate the model’s performance. While the high performance on the held-out PAMAP2 test set suggests that satisfactory results across multiple activities are achievable, other activity classifications in clinical populations remain unvalidated and require further research. That being said, it is important to clarify that the current study was primarily focused on developing a model to specifically identify walking, which allowed for greater emphasis on accurate walking labels. Moreover, one of the limitations surrounding many other models is that they are built on a single data collection source [18], whereas the current study demonstrated excellent results in identifying walking bouts across different datasets with different IMUs, placements, and populations, highlighting the robustness of the model. Additionally, this study did not fully validate the models on free-living data. Given the lack of a gold standard for activity classification in free-living environments, validation would have been challenging. Nonetheless, as noted previously, future studies will seek to assess the performance of these models in free-living data settings.

In summary, this study provides additional evidence that the ResNet + BiLSTM model is a valid approach to activity classification using IMU data. It also demonstrates the effectiveness of using open datasets, like PAMAP2, as a base that can be augmented with data from different sensor locations or populations. The ability to incorporate larger training datasets significantly improves model performance, as shown in this study. Furthermore, the deep learning model outperformed a heuristic frequency-based method when trained with data from a slower, pathological population. This is particularly relevant for future studies on free-living gait, as both healthy and clinical populations tend to walk slower and will require more robust gait identification.

## Figures and Tables

**Figure 1 sensors-25-06318-f001:**
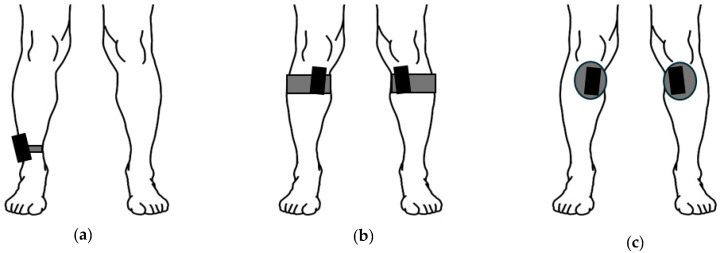
Sensor placements for PAMAP2 (**a**), healthy dataset (**b**), and clinical dataset (**c**). The PAMAP2 sensor was attached to the dominant side lateral ankle with a strap. The healthy dataset was attached using a semi-elastic strap, with the sensor placed medially and inferior to the knee. The clinical dataset used the same attachment location as the healthy dataset, but the sensors were attached using medical-grade tape. Please note that the IMUs are not to scale, as increased size is used to clearly show placements.

**Figure 2 sensors-25-06318-f002:**
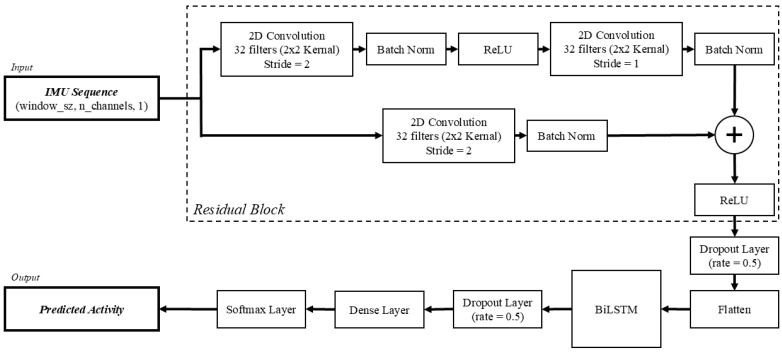
Model architecture of the ResNet + BiLSTM model, adapted from [17].

**Figure 3 sensors-25-06318-f003:**
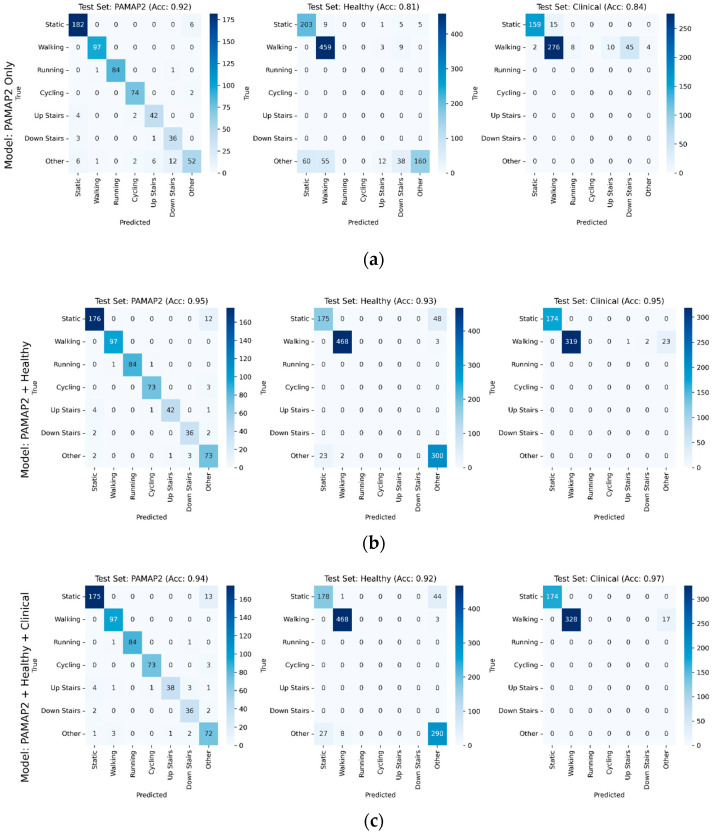
Confusion matrices broken out individually for PAMAP2, healthy, and clinical testing sets, with accuracy for each testing set shown for each model, for models using in training (**a**) PAMAP2 only, (**b**) PAMAP2 + Healthy datasets, and (**c**) PAMAP2 + Healthy + Clinical. Model performance generally improved on test sets as additional data was incorporated into training.

**Figure 4 sensors-25-06318-f004:**
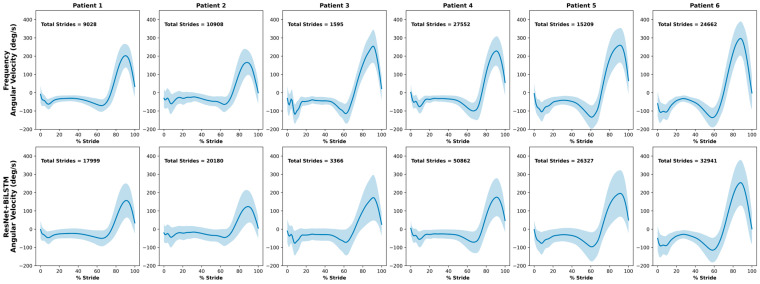
Average normalized mediolateral angular velocity of strides found during free-living for the clinical test set with the frequency-based method and the ResNet + BiLSTM model. Standard deviation is represented by the shaded region around each mean waveform.

**Table 1 sensors-25-06318-t001:** Description of each model iteration in terms of training, validation, and testing data used in model development and evaluation, with the number of n participants included from each dataset. Additional data was added for training and validation during model development, but the testing test was consistent across all model iterations.

Model Iteration	Training Data (n)	Validation Data (n)	Testing Data (n)
PAMAP2 Only	PAMAP2 (6)	PAMAP2 (1)	PAMAP2 (1) Healthy (2) Clinical (6)
PAMAP2 + Healthy	PAMAP2 (6) Healthy (10)	PAMAP2 (1) Healthy (2)	PAMAP2 (1) Healthy (2) Clinical (6)
PAMAP2 + Healthy + Clinical	PAMAP2 (6) Healthy (10) Clinical (26)	PAMAP2 (1) Healthy (2) Clinical (3)	PAMAP2 (1) Healthy (2) Clinical (6)

**Table 2 sensors-25-06318-t002:** Activity breakdown by percentage for each dataset, both individually and as a total breakdown by dataset combinations (i.e., PAMAP2 + healthy, PAMAP2 + healthy + clinical). Percentages are expressed in terms of the combined labeled activity data from the training, validation, and test sets.

Datasets	Static	Walking	Running	Cycling	Stair Ascent	Stair Descent	Other
PAMAP2	38.7%	16.3%	6.7%	11.2%	8.0%	7.2%	12.0%
Healthy	20.3%	53.6%	0.0%	0.0%	0.0%	0.0%	26.1%
Clinical	47.8%	52.2%	0.0%	0.0%	0.0%	0.0%	0.0%
PAMAP + Healthy	32.4%	29.0%	4.4%	7.4%	5.3%	4.7%	16.8%
PAMAP + Healthy + Clinical	36.0%	34.5%	3.4%	5.6%	4.0%	3.6%	12.8%

**Table 3 sensors-25-06318-t003:** Overall performance on training, validation, and test sets for each model.

Model Datasets	Performance Metric	PAMAP2	PAMAP2 + Healthy	PAMAP2 + Healthy + Clinical
Training	Accuracy	0.96	0.97	0.98
	Precision	0.96	0.97	0.98
	Recall	0.96	0.97	0.98
	F1-score	0.96	0.97	0.98
Validation	Accuracy	0.91	0.93	0.92
	Precision	0.93	0.93	0.92
	Recall	0.91	0.93	0.92
	F1-score	0.92	0.93	0.92
Testing	Accuracy	0.85	0.94	0.94
	Precision	0.89	0.94	0.94
	Recall	0.85	0.94	0.94
	F1-score	0.85	0.94	0.94

**Table 4 sensors-25-06318-t004:** Performance of the PAMAP2, healthy, and clinical trained model, stratified by activity class. The 95% confidence intervals were estimated using the Wilson score method.

Dataset	Class	Intervals	Accuracy	Precision	Recall	F1-Score
PAMAP2	Static	188	0.93 (0.89, 0.96)	0.97 (0.94, 0.99)	0.93 (0.89, 0.96)	0.95
	Walking	98	0.99 (0.94, 1.00)	0.98 (0.93, 0.99)	0.99 (0.94, 1.00)	0.98
	Running	86	0.98 (0.92, 0.99)	0.98 (0.92, 0.99)	0.98 (0.92, 0.99)	0.98
	Cycling	73	1.00 (0.95, 1.00)	1.00 (0.95, 1.00)	1.00 (0.95, 1.00)	1.00
	Upstairs	46	0.83 (0.69, 0.91)	1.00 (0.91, 1.00)	0.83 (0.69, 0.91)	0.90
	Downstairs	37	0.97 (0.86, 1.00)	0.92 (0.80, 0.97)	0.97 (0.86, 1.00)	0.95
	Other	77	0.94 (0.86, 0.97)	0.80 (0.71, 0.87)	0.94 (0.86, 0.97)	0.86
Healthy	Static	223	0.80 (0.74, 0.85)	0.87 (0.82, 0.91)	0.80 (0.74, 0.85)	0.83
	Walking	471	0.99 (0.98, 1.00)	1.00 (0.99, 1.00)	0.99 (0.98, 1.00)	1.00
	Other	317	0.91 (0.88, 0.94)	0.86 (0.82, 0.89)	0.91 (0.88, 0.94)	0.89
Clinic	Static	174	1.00 (0.98, 1.00)	1.00 (0.98, 1.00)	1.00 (0.98, 1.00)	1.00
	Walking	345	0.95 (0.92, 0.97)	1.00 (0.99, 1.00)	0.95 (0.92, 0.97)	0.97

**Table 5 sensors-25-06318-t005:** Performances of individual test sets from each dataset, based on the top performing model containing PAMAP2 + healthy + clinical datasets when used for gait detection and the heuristic frequency-based method. The 95% confidence intervals were estimated using the Wilson score method.

Dataset (n)	Model	Accuracy	Precision	Recall	F1-Score
PAMAP2 (1)	Frequency	0.79 (0.76–0.82)	0.40 (0.34–0.46)	0.99 (0.95–1.00)	0.57
	ML	0.99 (0.98–0.99)	0.95 (0.89–0.98)	0.97 (0.92–0.99)	0.96
Healthy (4)	Frequency	0.96 (0.95–0.97)	0.95 (0.92–0.96)	0.96 (0.94–0.98)	0.96
	ML	0.98 (0.98–0.99)	0.97 (0.95–0.98)	0.99 (0.98–1.00)	0.98
Clinic (6)	Frequency	0.87 (0.84–0.89)	0.96 (0.93–0.97)	0.84 (0.79–0.87)	0.89
	ML	0.96 (0.94–0.98)	1.00 (0.98–1.00)	0.95 (0.92–0.97)	0.97

**Table 6 sensors-25-06318-t006:** Method performance on participants in the clinical test set by gait speed. Individual performance metrics for each patient that are below 0.90 are shown in bold italics.

Patient	Gait Speed (m/s)	Speed Type	Method	Accuracy	Precision	Recall	F1-Score
1	0.30	Slow	Frequency	** *0.53* **	** *0.87* **	** *0.38* **	** *0.53* **
			ML	0.96	0.99	0.96	0.97
2	0.75	Slow	Frequency	** *0.81* **	1.00	** *0.72* **	** *0.84* **
			ML	0.93	1.00	** *0.89* **	0.94
3	1.00	Average	Frequency	0.96	0.99	0.94	0.97
			ML	0.97	0.96	0.99	0.98
4	0.99	Average	Frequency	0.95	0.99	0.94	0.97
			ML	0.98	0.99	0.98	0.99
5	1.32	Fast	Frequency	0.94	0.95	0.94	0.95
			ML	0.97	0.97	0.98	0.98
6	1.42	Fast	Frequency	0.98	1.00	0.96	0.98
			ML	0.99	0.98	0.99	0.99

## Data Availability

The PAMAP2 data utilized in this study are publicly available in the UC Irvine Machine Learning Repository at https://doi.org/10.24432/C5NW2H. The healthy and clinical datasets presented in this article are not readily available due to time limitations, while also subject to participants consent to their data being shared. Requests to access the datasets should be directed to Kim Madden.

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
