# Peer review of "Augmenting a ResNet + BiLSTM Deep Learning Model with Clinical Mobility Data Helps Outperform a Heuristic Frequency-Based Model for Walking Bout Segmentation"

_sensors, 2025, doi:10.3390/s25206318_

Round 1
Reviewer 1 Report
Comments and Suggestions for Authors
This paper augments a ResNet+BiLSTM deep learning model with clinical mobility data, and the results show that the 24 model trained with all three datasets performed best in terms of activity classification, 25 ultimately achieving a high accuracy of 96% on held out test data. This paper is well written and organized, but the following problems are required to be solved.
- More words about how to augments a ResNet+BiLSTM deep learning model should be provided.
- In line 103, this work…, in line 116, this study…, so it is suggested that the descriptions be unified, and the same applies to the following parts.
- More remarks should be provided to describe the Model architecture of the ResNet + BiLSTM model in Figure 2.
- The numbering of the formulas in lines 279-283 should be aligned.
- Does this model (ResNet + BiLSTM Model Framework) have any other application scenarios that can achieve excellent results?
- Some algorithms similar to those proposed in this paper are designed in the following literature: DOI: 10.1109/JIOT.2021.3067904, doi.org/10.1002/dac.989 and https://link.springer.com/article/10.1007/s11227-011-0708-z, Compared with the above algorithm, what are the advantages and disadvantages of this study?
Author Response
This paper augments a ResNet+BiLSTM deep learning model with clinical mobility data, and the results show that the 24 model trained with all three datasets performed best in terms of activity classification, 25 ultimately achieving a high accuracy of 96% on held out test data. This paper is well written and organized, but the following problems are required to be solved.
Response:
- We sincerely thank you for your time, effort, and thoughtful comments in reviewing our work. Your feedback has helped us to further improve the quality and clarity of the manuscript, and we believe we have addressed all concerns accordingly, as detailed below.
1. More words about how to augments a ResNet+BiLSTM deep learning model should be provided.
Response:
- Thank you for this comment. We agree that “augment” was unclear as to what we meant and have rewritten part of the introduction to better describe our approach (Lines 102-111). We also edited parts of the discussion to provide clarity as to what model was being augmented (i.e., the PAMAP2 base model described by Li and Wang; Lines 416-425).
2. In line 103, this work…, in line 116, this study…, so it is suggested that the descriptions be unified, and the same applies to the following parts.
Response:
- We have changed this to be consistent throughout to refer to the “study”.
3. More remarks should be provided to describe the Model architecture of the ResNet + BiLSTM model in Figure 2.
Response:
- Thank you for highlighting the need for additional clarification. We have added some additional details which we think will help clarify the model architecture (Lines 198-209; 217-236).
4. The numbering of the formulas in lines 279-283 should be aligned.
Response:
- We have aligned the numbering of the formulas.
5. Does this model (ResNet + BiLSTM Model Framework) have any other application scenarios that can achieve excellent results?
Response:
- Yes, there are quite a few different applications for this model framework, particularly in domains that have both spatial and temporal dependencies. We have added a sentence to highlight this in the discussion with the following references:
- Zhao et al., Math. Probl. Eng., 2018 — doi: 10.1155/2018/7316954
- Nafea et al., Sensors, 2021 — doi: 10.3390/s21062141
6. Some algorithms similar to those proposed in this paper are designed in the following literature: DOI: 10.1109/JIOT.2021.3067904, doi.org/10.1002/dac.989 and https://link.springer.com/article/10.1007/s11227-011-0708-z, Compared with the above algorithm, what are the advantages and disadvantages of this study?
Response:
- This is an interesting question. The three referenced papers primarily address system-level challenges that differ substantially from those targeted by our ResNet+BiLSTM framework, but they offer intriguing comparisons that we address below.
- Li et al. (2022): Their approach integrates deep reinforcement learning with genetic algorithms to optimize UAV routing and data collection in 6G IoT networks. Reinforcement learning excels in adaptive decision-making under uncertainty, which is advantageous for tasks such as route planning and dynamic resource allocation. However, RL models require carefully designed reward functions, substantial training data, and high computational overhead, which limit their direct applicability to wearable sensor–based activity classification. In contrast, our ResNet+BiLSTM is specifically designed for supervised learning on raw 3D accelerometer and gyroscope data, providing efficient extraction of spatial and temporal features for robust human activity recognition. While RL may be useful for optimizing sampling or adaptive model updating, it is not inherently suited to time-series signal interpretation in the same way as our framework.
- Lin et al. (2008): Their cross-layer WSN architecture leverages application-specific customization to improve energy efficiency and network performance. The advantage of such an approach lies in reduced redundancy and longer network lifetime, which is critical for large-scale or resource-constrained deployments. However, these gains come with potential disadvantages, such as added complexity in protocol design and sensitivity to idle-time prediction accuracy. By contrast, our ResNet+BiLSTM framework operates at the algorithmic level of signal interpretation, extracting meaningful activity patterns from sensor data. While WSN optimizations like Lin’s architecture can complement classification frameworks by improving data transmission efficiency, they do not themselves address the modeling of sequential sensor signals needed for activity recognition.
- Wang et al. (2011): Their CAGW_PD dynamic replication strategy improves distributed storage efficiency by replicating files based on weighted popularity measures, with proactive deletion to balance read latency and update overhead. This offers advantages for data availability and system responsiveness but is inherently tied to storage management. Potential disadvantages include increased computational overhead for managing access records and the risk of suboptimal replication decisions when popularity shifts rapidly. In contrast, our ResNet+BiLSTM model directly learns spatial-temporal representations from raw wearable sensor data, enabling classification rather than storage optimization. While adaptive weighting strategies like those in CAGW_PD could inspire future approaches for handling temporal imbalance or concept drift in sensor data, their current formulation is not designed for human activity recognition tasks.
- In summary, the cited methods provide valuable solutions in routing, network optimization, and storage management, but they operate at different layers of the data pipeline. Our ResNet+BiLSTM framework offers distinct algorithmic advantages for activity classification, including joint spatial-temporal feature extraction, robustness to gait variability, and demonstrated generalizability across healthy and clinical populations. We therefore view the cited methods as different, albeit with potentially complementary aspects, but ultimately distinct from the core problem addressed in this study. As such, they remain beyond the scope of our work and are only discussed here in response to the comment, rather than incorporated into the manuscript.

Reviewer 2 Report
Comments and Suggestions for Authors
This study demonstrates good performance across datasets. Have you considered cross-dataset cross-validation, for example, training on PAMAP2 and testing on clinical data, or vice versa? Are these results robust? Is there any significant performance degradation?
In the introduction, the authors should clearly state earlier the novelty of using clinical OA data to enhance the model, highlighting their contribution. In the methods section, the authors should further streamline the description of data preprocessing, including rescaling, windowing, and normalization, to avoid repetition. Furthermore, in the discussion section, the authors should strengthen the discussion of the model's generalization and clinical applicability, especially regarding potential challenges in free-living settings. Furthermore, in the core methods section, the authors should further discuss the model's misclassification of activities in the "other" category, which has a significant impact on the false positive rate in practical applications. Furthermore, the authors should provide a brief discussion of computational efficiency or real-time performance, especially regarding its feasibility for deployment on edge devices. Thanks.
Author Response
This study demonstrates good performance across datasets. Have you considered cross-dataset cross-validation, for example, training on PAMAP2 and testing on clinical data, or vice versa? Are these results robust? Is there any significant performance degradation?
Response:
- Thank you for your time and effort reviewing this study. We greatly appreciate your thoughtful feedback and have addressed your concerns in the responses below.
- We did evaluate multiple iterations of cross-dataset training and testing. For example, when the model was trained only on PAMAP2 and tested on the healthy and clinical datasets, it achieved reasonable performance with accuracies of at least 80% on the held-out sets. However, the additional datasets had important limitations, for instance, the clinical data contained only static and walking labels, which prevented full cross-dataset training and testing without discarding labels unique to the other datasets. By progressively layering datasets into training, we were able to leverage detailed datasets with limited participants alongside more sparsely labeled datasets, resulting in a framework that both aligned with the study aims and demonstrated robustness across diverse data sources.
In the introduction, the authors should clearly state earlier the novelty of using clinical OA data to enhance the model, highlighting their contribution.
Response:
- This is an excellent point. We have modified the introduction to reflect this (lines 107-111).
In the methods section, the authors should further streamline the description of data preprocessing, including rescaling, windowing, and normalization, to avoid repetition.
Response:
- Thank you for the suggestion. We agree that there was room to streamline and condense this section (i.e., 2.2 Data Processing) while removing repetitive statements and have revised this preprocessing section.
Furthermore, in the discussion section, the authors should strengthen the discussion of the model's generalization and clinical applicability, especially regarding potential challenges in free-living settings.
Response:
- We have added a paragraph regarding generalization and clinical applicability to the discussion section (Lines 510-527). Additionally, we have added new data, based on a similar comment by Reviewer 3. You will now notice that we have included out-of-lab data that provides further evidence for the superiority of this method (3.4 Effect of Gait Speed, Figure 4, Lines 405-408).
Furthermore, in the core methods section, the authors should further discuss the model's misclassification of activities in the "other" category, which has a significant impact on the false positive rate in practical applications.
Response:
- We have added a statement regarding the “other” category in the preprocessing section. (Lines 178-182).
Furthermore, the authors should provide a brief discussion of computational efficiency or real-time performance, especially regarding its feasibility for deployment on edge devices. Thanks.
Response:
- This is a great point that we have now addressed regarding generalizability of this model in the discussion (Lines 516-521):
- “With respect to gait, the model’s performance does seem to indicate an improvement over simpler heuristic models. However, a computationally intensive deep learning model like ResNet + BiLSTM may not be suitable for real-time feedback or devices with limited computational power. In these cases, the heuristic models may provide a practical implementation for identifying gait.”

Reviewer 3 Report
Comments and Suggestions for Authors
This manuscript addresses an important topic at the intersection of wearable sensing, machine learning, and clinical gait analysis.
While the manuscript presents a valuable contribution by validating deep learning methods in a clinical population, the novelty relative to Li and Wang (2022) should be more explicitly framed. The current study essentially retrains an existing architecture with additional datasets. The innovation lies more in the dataset augmentation and clinical testing rather than model design. This distinction should be emphasized in the Introduction and Discussion.
The clinical dataset includes only static and walking activities. This narrow activity scope may overestimate classification performance compared to more diverse real-world conditions. The authors acknowledge this in the Discussion, but a clearer statement of how this limitation might affect model generalizability is needed.
A central claim is improving walking bout detection for free-living applications, yet all data used were either laboratory-based or structured clinical assessments. Without validation on actual free-living data, claims regarding robustness in uncontrolled environments should be tempered or more cautiously phrased.
The frequency-based method serves as a useful baseline, but its implementation and parameter tuning are not described in full detail. For reproducibility, more transparency is needed about why specific thresholds were chosen and whether alternative parameterizations were tested.
While performance metrics are reported, no statistical testing is presented to assess whether observed differences between models are significant. Adding confidence intervals or statistical comparisons would strengthen the conclusions.
The clinical implications, particularly for monitoring patients with OA, should be expanded. For example, how might this model be integrated into rehabilitation or disease progression monitoring workflows? What advantages does it offer over existing wearable-based systems in practice?
Proofreading and consistency in terms.
Author Response
This manuscript addresses an important topic at the intersection of wearable sensing, machine learning, and clinical gait analysis.
Response:
- Thank you for your thoughtful and thorough review of our manuscript. We have addressed all of your concerns in the manuscript and responses below, and we believe these revisions have strengthened the paper.
While the manuscript presents a valuable contribution by validating deep learning methods in a clinical population, the novelty relative to Li and Wang (2022) should be more explicitly framed. The current study essentially retrains an existing architecture with additional datasets. The innovation lies more in the dataset augmentation and clinical testing rather than model design. This distinction should be emphasized in the Introduction and Discussion.
Response:
- Thank you for this suggestion. We agree that the primary novelty of this study lies in the application of the model to a clinical population and the augmentation of training datasets, rather than in the design of a new architecture. To emphasize this distinction, we have revised the closing paragraph of the Introduction (Lines 105–111) and highlighted this framing more explicitly in the Discussion (Lines 421–424; 454-457).
The clinical dataset includes only static and walking activities. This narrow activity scope may overestimate classification performance compared to more diverse real-world conditions. The authors acknowledge this in the Discussion, but a clearer statement of how this limitation might affect model generalizability is needed.
Response:
- We agree with this comment and do not wish to overstate the clinical results. In line with this and related feedback from other reviewers, we have added a paragraph in the Discussion explicitly addressing the limitations of the clinical dataset and how its narrow activity scope may affect model generalizability (Lines 511-513). We believe this addition sufficiently addresses the concern.
A central claim is improving walking bout detection for free-living applications, yet all data used were either laboratory-based or structured clinical assessments. Without validation on actual free-living data, claims regarding robustness in uncontrolled environments should be tempered or more cautiously phrased.
Response:
- This is a fair point. As noted in the original manuscript, clinical participants also wore sensors for up to a week in free-living conditions. While we chose not to include the full free-living dataset here given the scope of the manuscript, we have incorporated a qualitative example from the clinical test set to better support our statements regarding free-living utility, particularly for walking detection. Specifically, we added a plot of mediolateral angular velocity (reflecting swing phase) with strides identified by each method. The machine learning model detected approximately twice as many strides as the frequency-based method, and the angular velocity distribution was slightly broader, as expected. Importantly, the overall stride patterns were consistent across methods, indicating that the model not only captures more data but also maintains signal quality. Corresponding details have been added to the Methods, Results, and Discussion to clarify these additions.
The frequency-based method serves as a useful baseline, but its implementation and parameter tuning are not described in full detail. For reproducibility, more transparency is needed about why specific thresholds were chosen and whether alternative parameterizations were tested.
Response:
- This is an important point, and we have rewritten this part of the methods to better describe the pilot testing utilized to determine the peak prominence threshold. We believe that the updated text better reflects the analysis used to determine why the mediolateral angular velocity was used and the optimal threshold for peak prominence for this signal (Lines 279-300).
While performance metrics are reported, no statistical testing is presented to assess whether observed differences between models are significant. Adding confidence intervals or statistical comparisons would strengthen the conclusions.
Response:
- We appreciate the reviewer’s suggestion regarding the addition of confidence intervals or formal statistical comparisons. While we agree that such analyses can add interpretability, they are not commonly reported in the wearable gait analysis and activity recognition literature, where performance is often summarized using descriptive metrics such as accuracy, F1 score, and/or confusion matrices. To provide additional context and strengthen our conclusions, we have included results from free-living data, which offers a more ecologically valid assessment of model performance. Given the time constraints of this revision and the scope of the current manuscript, we were not able to compile confidence intervals or statistical comparisons for all model results. However, we agree that these could provide additional value, and if the reviewers feel this is critical for publication, we would be willing to address this in a subsequent revision.
The clinical implications, particularly for monitoring patients with OA, should be expanded. For example, how might this model be integrated into rehabilitation or disease progression monitoring workflows? What advantages does it offer over existing wearable-based systems in practice?
Response:
- We have added some additional statements around this within the paragraph regarding generalizability in the discussion section (Lines 510-527).
Proofreading and consistency in terms.
Response:
- In response to this and related feedback from other reviewers, we have carefully revised the manuscript to improve consistency in terminology throughout. Thank you again for your constructive comments.

Round 2
Reviewer 2 Report
Comments and Suggestions for Authors
The paper categorizes non-walking activities such as vacuuming and housework as "other," acknowledging that these activities may exhibit similar signal patterns to walking. This is a significant and under-discussed limitation in a real-world, free-living environment. Specifically, how many such "walking-like" daily activities will the model misclassify as "walking" in a week-long dataset of free-living activity? What is the magnitude of the systematic overestimation of clinical metrics like total steps and walking duration due to these false positives? The paper only demonstrates the model's performance on limited, labeled datasets (e.g., indoor walking), but severely lacks evaluation of its specificity in real-world, complex, and unconstrained environments—a critical aspect for its potential clinical application. Furthermore, some sections in the methods are repetitive, such as the descriptions of data preprocessing (windowing, normalization, etc.), which could be integrated for smoother flow. The discussion section logically explains the differences in model performance (e.g., compared to Li & Wang), but transitions between paragraphs could be smoother, particularly the connection between model performance and clinical implications. The explanation for the model's slightly lower performance on the PAMAP2 dataset (compared to Li & Wang) could be further elaborated, for example, regarding data distribution or label consistency. The discussion on computational efficiency (lines 516–521) is reasonable, but lacks specific solutions or lightweighting strategies, which could be expanded. Thanks.
Author Response
The paper categorizes non-walking activities such as vacuuming and housework as "other," acknowledging that these activities may exhibit similar signal patterns to walking. This is a significant and under-discussed limitation in a real-world, free-living environment. Specifically, how many such "walking-like" daily activities will the model misclassify as "walking" in a week-long dataset of free-living activity? What is the magnitude of the systematic overestimation of clinical metrics like total steps and walking duration due to these false positives? The paper only demonstrates the model's performance on limited, labeled datasets (e.g., indoor walking), but severely lacks evaluation of its specificity in real-world, complex, and unconstrained environments—a critical aspect for its potential clinical application.
Response:
- We thank the reviewer for raising this important point, which was one of the primary motivations of the present study. We agree that misclassification of “walking-like” daily activities is an under-discussed challenge for free-living gait analysis and is highly relevant for clinical applications. As the reviewer notes, our current datasets do not contain annotated free-living activity, and therefore we cannot directly quantify misclassification rates or the degree of potential overestimation of step counts and walking duration in unconstrained environments. While we added some qualitative analyses to illustrate deployment on free-living data in earlier revisions, addressing this question rigorously requires future work with ground-truth annotated free-living datasets. We have acknowledged this limitation explicitly in the manuscript (Lines 527–531). At the same time, our controlled dataset demonstrates that the approach can distinguish walking from other activities, suggesting that these issues may not substantially compromise performance, though further validation is needed.
Furthermore, some sections in the methods are repetitive, such as the descriptions of data preprocessing (windowing, normalization, etc.), which could be integrated for smoother flow.
Response:
- We appreciate the reviewer’s observation. In this revision, we have further condensed and streamlined the data preprocessing section (Section 2.2) to reduce redundancy and improve clarity.
The discussion section logically explains the differences in model performance (e.g., compared to Li & Wang), but transitions between paragraphs could be smoother, particularly the connection between model performance and clinical implications.
Response:
- We thank the reviewer for this helpful suggestion. In this revision, we have revised the discussion to improve transitions between paragraphs, particularly to strengthen the link between model performance and its potential clinical implications.
The explanation for the model's slightly lower performance on the PAMAP2 dataset (compared to Li & Wang) could be further elaborated, for example, regarding data distribution or label consistency.
Response:
- We thank the reviewer for this valuable suggestion. Upon reevaluating this section, we recognized that the original text contained redundant and potentially confusing details. We have rewritten the relevant portion of the discussion into a single, more concise paragraph that more clearly highlights the key factors influencing performance on the PAMAP2 dataset (Lines 431-447).
The discussion on computational efficiency (lines 516–521) is reasonable, but lacks specific solutions or lightweighting strategies, which could be expanded. Thanks.
Response:
- We have added the following text to address this important point (lines 502-505):
- “Future research could incorporate lightweighting strategies such as model pruning, quantization, or knowledge distillation to reduce computational costs while preserving performance. Alternatively, architectures with lower complexity (e.g. CNNs) could provide the efficiency necessary for on device deployment.”
Reviewer 3 Report
Comments and Suggestions for Authors
The manuscript presents an updated deep learning framework (ResNet+BiLSTM) trained on combined datasets, including clinical osteoarthritis (OA) populations, to improve walking bout segmentation from wearable IMU data.
While the manuscript acknowledges this limitation, the current evaluation is still heavily based on controlled or semi-structured tasks. Demonstrating performance in fully free-living, ecologically valid conditions would strengthen the claim of clinical applicability.
The clinical dataset only includes static and walking activities. As the model is intended for activity recognition, the restricted activity variety may inflate walking classification accuracy and limit generalizability.
The ResNet+BiLSTM framework is computationally demanding. Discussion of its suitability for real-time or wearable device implementation should be expanded, including possible model compression or lightweight alternatives.
The activity distribution across datasets may bias the model. It would be useful to report performance stratified by activity class imbalance.
The results are primarily descriptive. Confidence intervals or statistical tests comparing ML vs. heuristic methods would provide stronger support for performance claims.
Author Response
The manuscript presents an updated deep learning framework (ResNet+BiLSTM) trained on combined datasets, including clinical osteoarthritis (OA) populations, to improve walking bout segmentation from wearable IMU data.
Response:
- We thank the reviewer for their constructive feedback throughout the review process. We have carefully considered and incorporated their suggestions, and we believe all comments have now been addressed in the revised manuscript.
While the manuscript acknowledges this limitation, the current evaluation is still heavily based on controlled or semi-structured tasks. Demonstrating performance in fully free-living, ecologically valid conditions would strengthen the claim of clinical applicability.
Response:
- We appreciate the reviewer’s point and agree that validation in fully free-living, ecologically valid conditions is an important next step. In the previous revision, we added qualitative analyses using multi-day, free-living data from the clinical test set (Lines 404-414). However, as these datasets are not annotated, it is not possible to directly quantify model performance in this context. For this reason, we employed the frequency model as a comparator, providing side-by-side evaluation against a commonly used heuristic for gait detection. While this does not constitute true validation of the ResNet+BiLSTM model in free-living conditions, it offers an informative benchmark. We have also explicitly acknowledged the lack of free-living ground-truth validation as a limitation (Lines 527–531).
The clinical dataset only includes static and walking activities. As the model is intended for activity recognition, the restricted activity variety may inflate walking classification accuracy and limit generalizability.
Response:
- We thank the reviewer for highlighting this important consideration. We agree that the restricted activity variety in the clinical dataset may influence generalizability. To address this, we have previously included discussion of this issue in the context of generalizability (Lines 492-497) and explicitly noted it as a limitation (Lines 527–531). We therefore believe this concern has been adequately addressed in the current revision.
The ResNet+BiLSTM framework is computationally demanding. Discussion of its suitability for real-time or wearable device implementation should be expanded, including possible model compression or lightweight alternatives.
Response:
- We have added this to the discussion within the paragraph discussing generalizability (lines 502-505).
The activity distribution across datasets may bias the model. It would be useful to report performance stratified by activity class imbalance.
Response:
- Thank you for this suggestion. We have added a table (Table 4) in the results stratifying accuracy, with confidence intervals, with a brief description of these results (Lines 372-378)
The results are primarily descriptive. Confidence intervals or statistical tests comparing ML vs. heuristic methods would provide stronger support for performance claims.
Response:
- We thank the reviewer for this helpful suggestion. In response, we have added confidence intervals around classification accuracies using Wilson score intervals to provide additional context for the reported group results comparing the ML and frequency methods (Table 5). While the primary aim of this work was to compare machine learning and heuristic approaches across multiple datasets, the inclusion of confidence intervals strengthens the robustness of the performance claims and directly addresses the reviewer’s concern. Because the Wilson method required pooling the data, there were minor adjustments to the reported performance values (previously expressed as medians). These changes did not alter the overall conclusions or interpretation, but the relevant tables and text have been updated accordingly.